# Disulfiram Inhibits Opsonin-Independent Phagocytosis and Migration of Human Long-Lived In Vitro Cultured Phagocytes from Multiple Inflammatory Diseases

**DOI:** 10.3390/cells13060535

**Published:** 2024-03-18

**Authors:** Chen Li, Julian M. Schneider, E. Marion Schneider

**Affiliations:** Clinic for Anaesthesiology and Intensive Care Medicine, Ulm University Hospital, Albert-Einstein-Allee 23, 89081 Ulm, Germany; chenli072011@outlook.com (C.L.);

**Keywords:** long-lived in vitro cultured macrophages, dendritic cells, software to monitor phagocytosis and migration, disulfiram, mitochondria, metabolic activity, mitochondrial oxidative stress

## Abstract

Disulfiram (DSF), an anti-alcoholism medicine, exerts treatment effects in patients suffering from persistent Borreliosis and also exhibits anti-cancer effects through its copper chelating derivatives and induction of oxidative stress in mitochondria. Since chronic/persistent borreliosis is characterized by increased amounts of pro-inflammatory macrophages, this study investigated opsonin-independent phagocytosis, migration, and surface marker expression of in vivo activated and in vitro cultured human monocyte-derived phagocytes (macrophages and dendritic cells) with and without DSF treatment. Phagocytosis of non-opsonized Dynabeads^®^ M-450 and migration of macrophages and dendritic cells were monitored using live cell analyzer Juli™ Br for 24 h, imaging every 3.5 min. To simultaneously monitor phagocyte function, results were analyzed by a newly developed software based on the differential phase contrast images of cells before and after ingestion of Dynabeads. DSF decreased the phagocytic capacities exhibited by in vitro enriched and long-lived phagocytes. Although no chemotactic gradient was applied to the test system, vigorous spontaneous migration was observed. We therefore set up an algorithm to monitor and quantify both phagocytosis and migration simultaneously. DSF not only reduced phagocytosis in a majority of these long-lived phagocytes but also impaired their migration. Despite these selective effects by DSF, we found that DSF reduced the expression densities of surface antigens CD45 and CD14 in all of our long-lived phagocytes. In cells with a high metabolic activity and high mitochondrial contents, DSF led to cell death corresponding to mitochondrial oxidative stress, whereas metabolically inactive phagocytes survived our DSF treatment protocol. In conclusion, DSF affects the viability of metabolically active phagocytes by inducing mitochondrial stress and secondly attenuates phagocytosis and migration in some long-lived phagocytes.

## 1. Introduction

Macrophages and dendritic cells play a key role in sterile and infection-associated inflammation. Pro-inflammatory cytokines induced by such macrophages and dendritic cells explain a number of clinical features in persistent infections such as borreliosis [1,2]. In addition to cytokine secretions, macrophages and dendritic cells are active in the phagocytosis of infectious agents and migration to sites of infection and are responsible for the generation of local and systemic pain [1]. Generally, M1 macrophages are primary effectors of infection and inflammation [3]. However, these M1 macrophages may polarize to anti-inflammatory M2 macrophages, which counteract inflammation and improve tissue reconstitution [4]. This balance is severely impaired in conditions when pathogens are not appropriately eliminated from the body or when anti-inflammatory macrophages are overactivated, such as in the presence of malignancies [1]. The surface antigen CD38 [5,6] is responsible for immune cell activation and is accepted as a typical M1 marker, while CD163 is well-known as a typical M2 marker [7]. According to currently accepted knowledge, M1 and M2 macrophages differ by either using glycolysis or oxidative phosphorylation for their energy supply [8].

Various drugs in clinical use influence macrophage function, viability, phagocytosis, and migration [9,10]. Disulfiram (DSF) is a well-established drug used to treat alcoholism by inhibiting aldehyde dehydrogenase (ALDH) [11]. Moreover, DSF has been considered an anti-cancer agent by blocking the NF-kB pathway, increasing reactive oxygen species (ROS) formation, and inducing subsequent apoptosis in several types of human cancer [11]. In vitro studies have shown that DSF derivatives chelate zinc and copper [12], which explains their cytotoxicity to tumors. This property may also inhibit macrophage phagocytosis [13]. At the subcellular level, DSF affects mitochondrial membrane potential, depletes glutathione levels, oxidizes NAD(P)H, and inhibits oxidative phosphorylation [14]. Particles phagocytosed by macrophages are generally categorized as being opsonized or non-opsonized [15]. The uptake of opsonized particles, such as IgG-coated and complement (iC3b fragment)-coated particles [16], is considered conventional phagocytosis. In contrast, the uptake of non-opsonized particles, such as environmental dust, quartz, silica, and uncoated latex beads [17,18,19] follows the same requirements as the engulfment of Dynabeads^®^ M-450 Epoxy employed in this study.

The behavior of a certain cell population during an experiment can be observed using time-lapse recordings, essentially taking individual images of a defined visual field during microscopy every few minutes. The resulting video usually contains large amounts of data (movement, geometry, behavior of the cells). Although much of this information can be judged by the naked eye, automatized evaluation is highly desirable. Human interaction can be time-consuming and is also more prone to errors caused, for example, by fatigue or biased classification of the cells [20]. Digital image processing is of major interest in medical research. A number of mathematical formulas and algorithms have been developed to quantify and evaluate cell appearance and behavior in different microscopic scenarios. Using these methods, for example, with Matlab [21], cell movement and phagocytosis of different cell populations can be evaluated and quantified.

The current study focused on in vivo activated phagocytes (macrophages and dendritic cells), which were characterized as unique by extended lifetimes during in vitro culture. Such phagocytes were rarely enriched from healthy donors, but were a common feature in patients with infections and malignancies. In order to simultaneously quantify phagocytosis and migration, we set up an algorithm based on their video-recorded behavior. We then used DSF, a drug that is also used to treat borreliosis [22], and questioned whether DSF exhibited a special effect on such long-lived phagocytes for non-opsonized particle uptake and spontaneous migration.

## 2. Materials and Methods

### 2.1. Reagents

Disulfiram was purchased from Tocris (Banburry, Oxon, UK, www.tocris.com). Disulfiram at a concentration of 50 nM was dissolved in DMSO (Lab-Honeywell, Charlotte, NorCal, USA, www.lab-honeywell.com) with a final DMSO concentration of 0.01%. Dynabeads^®^ M-450 Epoxy were purchased from ThermoFisher (Rockford, IL, USA, www.thermofisher.com).

### 2.2. Cell Preparation

Human phagocytes were enriched by plastic adherence and subsequent in vitro culture using IMDM Iscove’s Modified Dulbecco’s Medium (www.bioscience.lonza.com, accessed on 1 October 2023), supplemented with Glutamax (thermofisher.com, accessed on 1 October 2023), antibiotics (8 µg/mL of gentamycin sulfate (www.merkgroup.com, accessed on 1 October 2023), 10 μg/mL BM cyclin (www.sigmaaldrich.com, accessed on 1 October 2023), and 10% endotoxin-free fetal calf serum (FCS superior, www.biosell.com accessed on 1 October 2023). EDTA-anti-coagulated blood samples from all patients underwent immune phenotype analysis using a panel of fluorescently labeled antibodies and flow cytometric analysis. Phagocytes from the following patients gave rise to long-lived phagocyte cultures and were subsequently studied: Chronic Borreliosis (*n* = 1), chronic fatigue syndrome (ME/CFS, *n* = 1), motor neuron disease (ALS, *n* = 1), pancreatic cancer (*n* = 1), treatment-resistant major depressive disorder (MDD, *n* = 1), post-coronavirus vaccination (postVAK syndrome, *n* = 1), active COVID-19 infection (*n* = 3), and glioblastoma °IV before surgery (GBM *n* = 3), approved by the Ethics Committee of Ulm University (U111-1179-3127, NCT02751138, 150/16, 82/07, 255/21, 98/2019, 400/2020). In detail, heparinized peripheral blood mononuclear cells (PBMCs) were separated using Ficoll gradient (www.biochrom.de, accessed on 1 October 2023) centrifugation. 6 × 10^5^ cells/mL PBMCs were seeded into plastic T25 tissue culture flasks and supplemented with 10% FCS, IMDM, at 37 °C in a humidified atmosphere with 5% CO_2_ overnight. Nonadherent cells were removed, and cultures continued in the absence of additional growth factors and cytokines. After 21–28 days of culturing, the remaining, mostly adherent fraction of monocyte-derived macrophages and dendritic cells were then treated with 50 nmol·L^−1^ of disulfiram (DSF) (www.tocris.com) or DMSO as a vehicle control (0.01% *v*/*v* final concentration) for 24 h at 37 °C in 5% CO_2_.

### 2.3. Macrophage Images Based on Phase–Contrast Microscopy

Morphological characteristics of the long-term cultured phagocytes were monitored by phase–contrast microscopy using a Nikon eclipse TS2 microscope and a 20×–40× phase contrast objective (Nikon, Tokyo, Japan, www.nikon.com, accessed on 1 October 2023). Images were taken using the NIS Elements Vs. 4.6 software (Nikon, Tokyo, Japan, www.nikon.com).

### 2.4. Cytospin Preparation and Wright–Giemsa Stain

Adherent phagocytes were detached by incubating the adherent cell layer in PBS (Phosphate-Buffered Saline) with 0.002% of EDTA at room temperature, washed with PBS (containing 0.5% FCS), and resuspended at a final concentration of 5 × 10^5^ cell/mL. Glass microscope slides were labeled, placed under the cardboard filter, and assembled with the sample chambers into appropriate slots in the balanced cytospin centrifuge (Shandon.com). A cell suspension of 100–200 μL was pipetted into each sample chamber and spun at 300 g for 5 min. Slides with flattened and dispersed cells were dried overnight and subsequently stained by Wright’s Giemsa staining using the Hemocolor^®^ solutions 1-3. Images were taken with a 60× objective (Nikon TS2 inverted microscope, Nikon, Tokyo, Japan, www.nikon.com).

### 2.5. Monitoring Phagocytosis by Video Microscopy

The phagocytes cultured in T-25 culture flasks were counted by phase contrast microscopy using NIS Elements Vs. 4.6 software (nikon.com), then treated with either disulfiram (DSF) of 50 nmol·L^−1^ or DMSO as a vehicle control (0.01% *v*/*v* final concentration), and an equivalent of 20 Dynabeads^®^ M-450 Epoxy were added per macrophage. Immediately after adding the Dynabeads, cell layers in the T-25 flasks were continuously recorded by video microscopy using a Juli™ Br (Live cell analyzer, Julabo, Seelbach, BY, Germany, www.julabo.com) incubator microscope equipped with a 4× objective. A time-lapse sequence was recorded with a frame rate of one image per 3.5 min for 24 h at 37 °C in a humidified atmosphere.

For cell movement analysis, a multiple-object tracking method implemented in MatLab Vs. R2020a (The MathWorks, Natick, MA, USA, www.mathworks.com) was used to track the Dynabead conglomerates that formed after ingestion of the phagocytes. A review and, if necessary, correction of the tracks were performed by a human operator. Centroids of the detected conglomerates were used as an approximation for the position of the corresponding cell.

For quantification of phagocytosis in the time-lapse videos, a custom pattern recognition and classification method was implemented in MatLab Vs. R2020a. This algorithm detects conglomerates of beads and uses image analysis tools to determine (i) the quantity of beads in the conglomerate and (ii) whether the conglomerate had been engulfed by a phagocyte or was represented extracellularly.

### 2.6. Metabolic Characterization Cultured Macrophages and Dendritic Cells Activated In Vivo

The supernatant of cell culture was collected from in vitro cultured dendritic cells and macrophages before and after treatment with 50 nmol·L^−1^ DSF, or DMSO as a vehicle control (0.01% *v*/*v* final concentration) for 24 h. Eventual cell debris from the supernatant was removed by centrifugation before measurement. Levels of glucose and lactate were detected automatically by an ABL8000 FLEX blood gas analyzer (Raiometer, Copenhagen, Denmark, www.radiometer.de), according to the manufacturer’s protocol.

### 2.7. Flow Cytometry

For surface phenotypic markers assessment, macrophages were stained with fluorophore-labeled extracellular antibodies IgG1 (bioclone.com), IgG2a (bioclone.com) CD45 (bdbiosciences.com), CD14 (bdbiosciences.com), CD163 (miltenyibiotec.com), CD38 (bdbiosciences.com), and HLA-DR (clone L243, biolegend.com) for 30 min protected from light on ice, washed with PBS (thermofisher.com) containing 0.25% FCS (biochrom.com) and 0.1% sodium azide (www.merkgroup.com), and measured by flow cytometry. For the analysis of mitochondrial contents, we used Mitotracker^TM^ (thermofisher.com), and for the quantification of mitochondrial ROS analysis, we used Mitosox^TM^ (sigmaaldrich.com). Phagocytes were detached from culture flasks, washed with prewarmed Hank’s Balanced salt solution (HBSS) (thermofisher.com), and incubated with 500 nM Mitotracker^TM^ or 5 μM MitoSOX^TM^ in HBSS at 37 °C and 5% CO_2_ for 20 min. Then, cells were washed with prewarmed HBSS and immediately analyzed by flow cytometry. For the determination of cell death, propidium iodide (sigmaaldrich.com) (0.1 mg/mL) and flow cytometry were implemented. At least 5 × 10^3^ cell events were collected for measurement by flow cytometry. The forward scatter (FSC) and the side scatter (SSC) parameters were used to reflect cell size and cell complexity, respectively. Emission of antibody-labeled phycoerythrin (PE) and fluorescein isothiocyanate (FITC) were detected at 575 nm and 525 nm, respectively. Measurement was performed using a FACSCalibur™ device (BD biosciences, Franklin L., NJ, USA, bdbiosciences.com). The FACSCalibur™ instrument and CellQuest™ software Vs. 5 (bdbiosciences.com) were used for flow cytometric analysis. Mean fluorescence intensities were calculated as follows: [%positive gated for marker] ∗ [axis mean of positive gated quadrants for marker] − [%positive gated for isotype of marker antibody] ∗ [axis mean of positive gated quadrants for isotype of marker antibody].

### 2.8. Statistics

All data analyses and graphical representations were performed using GraphPadPrism Vs. 9.1.1 (www.graphpad.com). An unpaired two-tailed *t*-test was used for comparing data between two groups within the same time point. Differences with *p* < 0.05 were considered to be significant; differences with *p* < 0.001 were considered to be very significant.

## 3. Results

### 3.1. Quantification of Functional Phenotypes Expressed by Cultured Macrophages

#### 3.1.1. Phagocytosis

Cultured phagocytes were subjected to live video microscopy using the Juli™ Br device. For quantification of phagocytosis and visually improved detection, Dynabeads^®^ M-450 Epoxy were added to cell cultures at a concentration ratio of 1 cell to 15–20 DynaBeads. Following bead incubation, phagocytes were subjected to video recording for their velocity and efficacy of bead uptake, as well as their migration speed. The algorithm applied for phagocytosis activity operates as follows: first, all conglomerated DynaBeads were counted in an image using brightness (cells) and contrast values (for DynaBeads) combined with thresholds in size; second, classification of any given detection into free and ingested conglomerates was performed and color coded (Figure 1). For the distinction of Dynabead conglomerates and ingested Dynabeads by active phagocytes, the following parameters were taken into account: (i) size of the conglomerate; (ii) position of the conglomerate using a mask that was generated from the movement analysis data to estimate the cell positions. Conglomerates coincident with the position of a cell were classified as ingested; (iii) image contrast of adjacent pixels (cell borders show stronger contrasts than empty spaces, a conglomerate with low contrast in its surrounding is likely a free Dynal Bead); (iv) counting of beads in all detected free conglomerates (and sorting out artifacts) was achieved as follows: (i) creation of a size-brightness diagram for all detections; (ii) hotspots of the diagram corresponding to conglomerates of one, two, three, etc. Dynabeads, respectively; (iii) outliers with large sizes but similar brightness corresponded to larger conglomerates in which ingested bead numbers could be estimated by phase contrast density and cell size. Summing up all cell-free beads and conglomerate numbers, the program returns the number of free beads in every image. Assuming the experiment started without any ingested beads, subtracting this number from a bead count of the first image of the experiment yielded the total number of ingested beads. Resulting numbers were post-processed with smoothing filters to reduce noise and to allow normalization by the number of beads and cells in the defined visual field of the entire video. (The full video recordings of the two phagocyte cultures shown in Figure 1 are available as Appendix A).

#### 3.1.2. Cell Movement Analysis for Migration Distance Velocity

For cell movement analysis, three types of migration paths were identified from data based on the method ‘multiple-object tracking’ implemented in our software for every cultured macrophage: (i) the (raw) path as a sequence of the cell’s centroids between frames, (ii) a smoothed path of the cell using a robust linear regression (20 frames window), omitting stationary oscillations of the cell and reducing the influence of noisy centroid detections, and (iii) the effective path of the cell between start and end of its path, i.e., its endpoint minus its starting point, named Euclidian distance [23] traveled by an individual macrophage (Figure 2). Mean cell velocities for the three path options were derived from all cells in a given visual frame. High velocities, as exemplified in Figure 2a, represent strong oscillations, while high velocities in Figure 2b correspond to movement in a defined direction of the cell. The velocity derived from path analysis in Figure 2c implies chemokinetic effects, which is related to the final Euclidian distance.

### 3.2. Identification of Macrophage Phenotypes

Before investigating the potential mechanism of opsonin-independent phagocytosis regulated by DSF, we first identified the phenotypes of cultured phagocytes derived from patients with their respective diagnoses using flow cytometry. In addition, age, gender, and the presence or absence of exosomes are shown (Table 1). Exosomes determined by flow cytometry were identified by the expression of CD63 but absence of the leukocyte-specific marker CD45. In all CD45-expressing cell cultures, the phenotype with negative expression of CD14 was tentatively assigned as a dendritic cell, being either cDC (conventional DC) or iDC (immature DC) [24], and those with positive expression of Kv1.3 were defined as mature and activated phagocytes [25]. Dendritic cells with coexpression of CD163, CD141, and low CD14 expression densities were designated as DC-10 [26]. Phenotypes with positive expression of CD38 were defined as M1 [5,6]. Macrophages exhibiting only high expression of CD163 were defined as M2c [27,28]. Macrophages with a combination of CD163 and CD206 expression were defined as M2a [27]. Phagocyte cultures bearing phenotypically different macrophages and dendritic cells are marked by their relative percentages. The phenotypic analysis of each long-lived cell culture is displayed in Table 2.

### 3.3. Modulation of Macrophage and Dendritic Cell Function and Viability by DSF

#### 3.3.1. Correlation of Reduced Phagocytosis and Cell Death Induced by DSF

Using the developed software and algorithm (Figure 1), we observed a significant difference in the phagocytosis of non-opsonized Dynabeads^®^ M-450 Epoxy between untreated and DSF-treated phagocytes. As shown in Figure 3A,B, DSF treatment led to a reduction of cell densities in a classical CD14-negative DC (#20231) phagocyte culture. In addition, the number of beads ingested by phagocytes was diminished (84 cells decreased to 54 cells in a video frame). Migration speed (smoothed path) and chemokinesis were significantly affected in individual cells of the phagocyte culture (*p* < 0.001; *p* < 0.05) (Figure 3). But, when we monitored the percentage of beads ingested per cell, DSF did not result in a significant inhibition. These results show that #20231 phagocytes are affected in cell movement and not so much in bead ingestion when 50 nM of DSF were applied. In contrast, the phagocyte culture #20315 with iDC phenotype appeared to be more sensitive to DSF treatment with regard to cell numbers (58 cells decreased to 16 cells in a video frame), migration, migration speed, and chemokinesis (Figure 4). In addition, the number of phagocytes that could be recorded was remarkably lower in DSF-treated cultures. Interestingly, the increased DSF concentration of 100 nM compared to 50 nM DSF resulted in a comparable effect. Observation of phagocytosis over a time period of 24 h showed almost full inhibition of the phagocytic activity over time with both DSF concentrations (Figure 4).

The observation shown in Figure 3 was confirmed by the analysis of four other phagocyte cultures (one DC culture from another GBM patient #20331 with cDC phenotype; one M2a culture from a patient of chronic borreliosis #20725; one DC culture from a patient with major depressive disorder #21161; one M2a culture from a patient suffering from a post-vaccination disease (post-VAC syndrome) #21164) (Figure 5). The results of these experiments further showed that the cells with the highest phagocytic capacity were affected most by DSF treatment (Figure 6).

In cell cultures that were significantly affected by DSF during recording, morphological examination showed cell shrinkage as well as vacuolization of the cytoplasm, which might indicate endoplasmic reticulum-(ER-) stress (Figure 7, red arrow) and occasional cell blebbing (Figure 7, blue arrow).

#### 3.3.2. Correlation of DSF-Induced Mitochondrial ROS and Cell Death

We then questioned whether changes in phagocytic capacity and viability observed in individual phagocyte cultures by DSF treatment were due to changes in mitochondrial membrane potential and/or mitochondrial oxygen radical formation. Mitotracker^®^ Red was applied to quantify the mitochondrial mass and mitochondrial membrane potential; MitoSOX^®^ Red was used as a mitochondrial superoxide indicator; and propidium iodide (PI) was used to measure cell death. As shown in Figure 8A, the mitochondrial mass of cultured M1 and M2a polarization was selectively decreased by DSF. In addition, mitochondrial ROS formation was increased, with the exception of cDC (#21161 and #21163) (Figure 8B). DSF-induced cell death was high in three M2a phagocytes (#20373, #21164, and #20705) and one M1 (#20643). A smaller effect on cell viability was detected in M2c (#21162). However, DSF did not influence the viability of DC phagocytes (#20650, #21161, #21164) (Figure 8C). In this case, DSF-induced cell death appeared to correlate with mitochondrial ROS formation, preferentially in M2a polarized cells. M2a cells relying on mitochondrial oxidative phosphorylation were more susceptible to the cytotoxic effect of DSF.

Additionally, we asked whether the reduction of mitotracker staining was due to the release or extrusion of mitochondria from DSF-treated phagocytes. As shown in the representative population of untreated phagocytes gated in Appendix A, exosomes contained mitochondrial mass that likely originated from cultured phagocytes. As shown in Appendix A, the number of isolated mitochondria in DSF-treated phagocytes was not different from non-treated phagocytes. Likewise, exosomes also contained a small qauntity of ROS generated by mitochondria, particularly in phagocytes with M2 phenotypes derived from the same two COVID-19 patients, #20373 and #20705. However, DSF played a heterogeneous role in the regulation of mitochondrial ROS levels in exosomes’ population. These results suggest that the exosomes’ fraction detected by flow cytometry contained mitochondria, but the effect of DSF on mitochondrial function of phagocytes was apparently independent from mitochondrial fragments in these exosomes.

#### 3.3.3. Correlation between Glycolytic Metabolism and Phagocyte Phenotype Regulated by DSF

Since phagocytes’ polarization and activities are specifically determined by their metabolic processes, we evaluated the glycolytic level of long-lived phagocytes. Glucose, the fuel consumed in glycolysis, and lactate, the inevitable end product of glycolysis, were assessed. As shown in Figure 9, the level of glucose consumption was approximately proportional to that of lactate generation in each cell culture. However, one macrophage culture with M2a #20373 and cDC #21163 unexpectedly showed the highest consumption of glucose and production of lactate when compared with M1 and iDC phenotypes. Accordingly, the phenotypes of cultured phagocytes were not strictly reflected by the levels of indicative metabolism involved in phenotypic phagocytes, which might be impacted by the in vivo activated immune response, e.g., the micromilieu of COVID-19.

Further, we classified our cultured phagocytes into two groups, with low and high levels of glycolysis, respectively, by the cutoff values of glucose consumption 80 mg/dL and lactate production 7.6 mmol/L. As shown in Appendix A, in phagocyte cultures with either high or low levels of glycolysis, DSF did not show a significant influence on glucose consumption and lactate production, with the exception of two patients with long-COVID: DSF reduced the glucose consumption of #20705-M2a by 19 mg/dL to 49 mg/dL while increasing the lactate production of #21162-M2c by 4.3 mmol/L to 9.2 mmol/L.

#### 3.3.4. Modulation of Surface Antigen Expression by DSF

The effect of DSF treatment on surface marker expression is shown in Figure 10. The leukocyte common antigens CD45 and CD14 were reduced upon DSF treatment. The densities of these antigens were generally higher in phagocytes with lower glucose consumption, and the effect by DSF was more pronounced. However, the expression of human leukocyte antigen (HLA-DR) was not altered by DSF (Figure 10).

## 4. Discussion

This study used a novel algorithm and image analysis to follow the migration and phagocytosis of exceptionally long-lived phagocytes from patients and a healthy donor, enriched by long-term in vitro cultures in the absence of exogenous growth factors. In addition, we questioned whether DSF, an inhibitor of ALDH and chemotherapeutic agent [29,30], would selectively affect the function, viability, and phenotype of these long-lived phagocytes. Long-lived phagocytes were either macrophages or dendritic cells. Macrophage and dendritic cells associated with phagocytosis and migration were tested by video–microscopy. DSF generally resulted in reduced phagocytosis and the migration of both velocity and migration to different extents. These results may correspond to findings linked to the blockade of SOCS3, a transcription factor [23]. The reduced capacity for phagocytosis was most pronounced by the effect on phagocytes with high phagocytic activity, suggesting a significant effect on the cytoskeleton and velocity of particle uptake, since phagocytes ingesting more than 10 beads/cells move significantly faster to reach Dynabeads for phagocytosis. Phagocytosis is a highly energy-consuming process [31], and therefore, the reduced phagocytosis mediated by DSF was linked to lower amounts of mitochondria as well as increased mitochondrial stress. Increased oxidative stress induced by DSF has been previously described in the context of DSF serving as an anti-neoplastic agent [11].

Phagocytosis is a specialized characteristic of phagocytes, as well as tumor cells, within the complex network of cellular immunity. Phagocytosis requires membrane rearrangement and is physiologically mediated by specific receptors [16]. To explore the effect of DSF on phagocytosis, migration, and chemokinesis of individual phagocytes, we found that DSF downmodulated phagocytosis and migrations, likely due to chemokinesis, in both DC and macrophage subtypes of individual patients. However, DSF-induced cytotoxicity was restricted to phagocytes with high mitochondrial contents and was more constantly found in cultures with macrophages phenotypes (M1, M2a, M2c) but not DC phenotype (Figure 8A,C). Migration velocities and migration paths were also influenced, which may indicate a DSF effect on phagocytes. This effect is likely due to the cytoskeletal network and would also explain the reduced expression densities of CD45 and CD14. However, HL-DR expression was not altered by DSF treatment. Thus, HLA-DR trafficking appears to underlie a different trafficking control when compared to CD45 and CD14. In this study, we applied non-opsonized Dynabeads^®^ M-450 Epoxy to monitor and quantify ingestion by phagocytes. Consecutively, rapid ingestion of Dynabeads^®^ M-450 Epoxy by the majority of cultured human long-lived phagocytes can be observed. Nevertheless, the hydrophobic residues of Dynabeads^®^ M-450 Epoxy can bind to amino residues of proteins, which may cause a high affinity for cell membrane components, such as cholesterol and phospholipids, in the culture medium at neutral pH [32]. On the other hand, Gu and colleagues found that the inhibitory effect of serum on non-opsonized particle uptake was determined by the negatively charged serum glycoproteins [33], which constitute the main proportion of serum nutrients derived from bovine sources [34]. Since our culture conditions included 10% fetal calf serum, we also performed Dynabead phagocytosis experiments in the absence of fetal calf serum and in the presence of cyclodextrin, which should scavenge phospholipids for opsonization in serum-containing medium. Results showed that in the absence of serum, phagocytosis for Dynabeads by phagocytes was decreased (Appendix A).

When analyzing toxicities by DSF, we found that phagocyte cultures with high mitochondrial contents showed upregulation of ROS in their mitochondria upon DSF treatment, followed by higher levels of cell death. In addition, Mitotracker^®^ Red showed that DSF selectively impaired mitochondrial mass and their membrane potential. Mitotracker^®^ Red is a cell-permeable dye that binds to thiol-reactive chloromethyl groups in the mitochondial membrane [35]. This fluorescent dye reflects mitochondrial mass and metabolic activity levels in live cells. High fluorescent intensity of Mitotracker^®^ Red manifests a large amount of vital cellular mitochondrion based on mitochondrial membrane potential [35]. Concomitantly, ROS formation in mitochondria was determined by MitoSOX^®^ Red. This probe is a mitochondrial superoxide indicator reflecting the level of cellular oxidative stress caused by reactive oxygen species (ROS), which are generated by mitochondria and also play an essential role in cell damage [36]. As one of the regulators in phagocytosis, intracellular ROS produced by phagocytes themselves relies on NADPH-Oxidase activity (NOX2). Increased ROS levels are involved in the phagocytic process [37]. In this study, the DSF-mediated increase in mitochondrial ROS was found in phagocyte cultures when DSF also impaired the phagocytic function most. This could be a reflection of the impairment on mitochondrial function affecting the physiological balance. DSF also decreased the surface expression of the leukocyte common antigen CD45, a phosphatase that regulates cell proliferation, migration, and immunity [38,39]. We propose that DSF leads to energy depletion, impacting intracellular trafficking of surface receptors [40]. In addition, CD14 expression was also reduced. CDF14 plays a significant role in the recognition of LPS and subsequent pathogen-induced immunity in phagocytes. Further experiments are necessary to decide whether DSF affects CD14 membrane expression or leads to increased release from the cell surface. Remarkably, HLA-DR expression densities were not affected by DSF. For HLA-DR detection, we applied the L243 antibody, which detects HLA-DR, including the MHC i-chain bound HLA-class II molecules as well [41]. These results indicate that the trafficking of membrane proteins may underlie different molecular mechanisms [42]. The insensitivity of HLA-DR to DSF is of high relevance due to its importance in immune regulation [43].

Recently, the effect of DSF has been attributed to its metabolic compounds arising from cytochrome p450 2E1 metabolism [44]. The polymorphic properties of Cyp450 2E1 may easily explain variations in DSF efficacy, but these may be equally due to the phagocyte subtype. According to the literature, M1 macrophages have been described with low mitochondrial contents, and M2 macrophages have been described with high mitochondrial contents [45]. Howevwer, our in vivo activated and in vitro enriched long-lived phagocytes do not follow the polarization schedule induced by exogenous cytokine supplements. Moreover, neither low mitochondrial contents nor high glycolytic activity and lactate production have been found in the one M1 macrophage culture derived from a healthy donor. In general, M1 macrophages rely on glycolysis for ATP generation [46], and M2 macrophages should display high levels of mitochondrial oxidative phosphorylation [47]. The current study shows that the mitochondrial and metabolic characteristics of M1/M2 phenotypes do not harmonize with the properties of our long-lived phagocyte cultures. Nevertheless, DSF-induced toxicities and cell death were related to mitochondrial contents in our phagocyte cultures, apparently independent of macrophage and dendritic cell subtypes. However, in the current study, glycolysis was not observed as a specific feature of the M1 phenotype. In contrast, it was found to be more strongly performed in macrophages of the M2 phenotype and dendritic cells when compared with M1 (Figure 9). These results might be in accordance with the evidence that glycolysis could be inhibited by the preventive machinery from glucose depletion; on the other hand, glycolysis could be essential for M2 activation [48].

In addition, DSF has been identified as a potent PLpro inhibitor and MPro inhibitor of SARS-CoV2 [49,50,51]. Further, disulfiram has been described to exhibit antiviral activity in cell culture, targeting nsp13,m nsp14 of SARS-CoV2 and inhibiting pyroptosis by gasdermin D inhibition [52,53]. These observations support the current analysis focusing on phagocytes’ function and could provide potential evidence to explain the higher sensitivity of cultured macrophages derived from COVID-19 patients in response to DSF-induced effects (Figure 8B,C, Appendix A). Moreover, disulfiram has also been used as an antimicrobial to treat Lyme borreliosis for decades. It alleviated the pathogenicity of Lyme borreliosis by inhibiting the growth of B. burgdorferi (Bb), reducing the pro-inflammatory cytokines, e.g., TNF, IL-1β, and IFNγ, the levels of Bb-specific immunoglobulins, e.g., IgM and IgG, as well as the ratio of activated leukocytes, e.g., CD19^+^ B cells, whereas it increased CD3^+^CD4^+^, naïve, effector, and memory T cells for bacteria elimination [54]. Disulfiram was also identified to inhibit B. burgdorferi growth by its high affinity for metal ions zinc and manganese, which are necessary for the spirochaetes’ metabolism [55]. Likewise, disulfiram combined with zinc enhanced the toxicity of zinc on macrophage phagocytosis [13], because diethyldithiocarbamate (DDTC), a unique metabolite of disulfiram, played a crucial role in chelating metal ions, e.g., copper and zinc, as well as metal-containing enzymes that are indispensable in cell activities [56].

Monocytes, macrophages, and dendritic cells share many features, including some phenotypic markers [57]. In vivo, monocytes originate from hematopoietic stem cells (HSCs) and differentiate into tissue-resident macrophages or dendritic cells (DCs) when they encounter danger and pathogen-derived signals in the circulation [58]. The in vitro differentiation of monocytes should mimic the model for the in vivo process by monocytes’ maturation. Having this concept in mind, we cultured the plastic-adherent fraction of antigen-presenting cells from blood samples in vitro, without a selective differentiation stimulus. Our aim was to investigate the closest phenotype of the monocytes from blood donors of patients. For the characterization of monocyte-derived macrophages and dendritic cells, we here applied flow cytometry and selectively included markers defining classical CD14^−^ dendritic cell (DC) [59], M1- and M2-specific antigen expression [60,61], and subclassified dendritic cells based on low expression density of CD14, its complete absence, and the co-expression of CD141, CD209, CD206, and CD163. Pre-activation states were defined as fully activated and matured macrophage subtypes, M1 and M2. The expression of CD163 and CD38 was found to distinguish M1 vs. M2 polarized macrophage cultures best. CD38 expression densities are related to immune activation and were recognized to distinguish M1 form the other phenotypes [62]. In our data, we found a significant decrease in CD38 expression when we compared fresh monocytes with long-term cultured phagocytes. However, the higher fluorescence intensity of CD38 in monocytes in vivo also corresponded to an M1 phenotype in cultured macrophages, indicating that CD38 is indeed a reliable M1 marker for investigation. CD38 is a special enzyme, responsible for the conversion of NAD to cyclic ADP-ribose (cADPR) and the activation of Ca^+^ signaling, which plays a crucial role in inflammatory pathways [63]. The CD38 antigen provides signal induction for T- and B- lymphocyte activation [64]. In contrast, CD163 levels of some macrophage cultures experienced an increase or decrease during in vitro culture. The scavenger receptor CD163 expression is restricted to the monocyte/macrophage lineage [65]. Increased CD163 expression defines the anti-inflammatory phase in which immune cells may counter-regulate inflammation [66]. As such, a higher level of CD163 in dendritic cell cultures of glioblastoma patients may reflect an inflammatory property. More markers need to be tested to subclassify dendritic cells related to a tumor environment [2].

Limitations of this study could be summarized as follows: automated video microscopy analysis was classified into automated cell movement analysis in addition to automated phagocytosis analysis. For cells on their own, the images provided by the live cell analyzer Juli™ Br lack the necessary phase contrast and detail to implement a reliable detection and tracking algorithm. As a workaround, detection and tracking was performed on conglomerates of Dynabeads, which in most cases coincide with phagocyting cells. Erroneous detections (e.g., free conglomerates) were excluded by a series of filters, and results were verified by a human operator in a later step. Apart from detecting noise, this procedure facilitates a reliable movement analysis for phagocyting cells; however, non-phagocyting cells were not analyzed with this approach and require a setup with better optics. The smoothing of the cell path (Figure 2b) is necessary to eliminate the noise caused, e.g., by small oscillations of the cell. The mathematical parameters and type of this smoothing were determined heuristically. While these smoothed paths yielded a more realistic image of the cells’ migration, the heuristic approach makes it difficult to directly compare migration velocities to different research work simply by the ‘multiple-object tracking’ method. Regarding the limitations of automated phagocytosis analysis, in order to determine if a specific cell conglomerate has been ingested by a cell or not, the automated algorithm tries to look for indications of a present cell around the conglomerate. As mentioned above, cell contrast in the live cell analyzer Juli™ Br images is scarce. In order to enhance detection accuracy, the algorithm was also provided data from the cell movement analyses to aid cell detection. After the analysis, a human operator evaluated the results for probability and adjusted detection parameters if necessary. A smoothing filter further reduced the influence of erroneous detections. While the live cell analyzer Juli™ Br guarantees comparable conditions during multiple experiments, the amount of cells and Dynabeads in every experiment varies, introducing statistical dependencies. To try to account for this in the phagocytosis analyses, the amount of beads ingested was normalized to the amount of cells and/or the amount of Dynabeads present in the respective experiment. To better discriminate the effects of DSF on the viability of activated macrophages and dendritic cells, another set of experiments will be necessary, including activation of these cultures with pattern recognition receptors, such as Toll-like ligands and pathogens, especially those involved in borreliosis.

## Figures and Tables

**Figure 1 cells-13-00535-f001:**
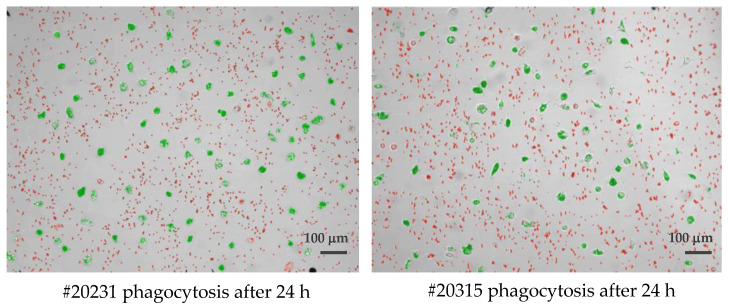
Single frame of two long-lived phagocytes (#20231 **left** and #20315 **right**), demonstrating the classification of detected beads being identified as either cell-free agglomerates (colored red) or engulfed/intracellular beads (colored green) by cultured macrophages after 24 h of observation. Scale bar: 100 μm.

**Figure 2 cells-13-00535-f002:**
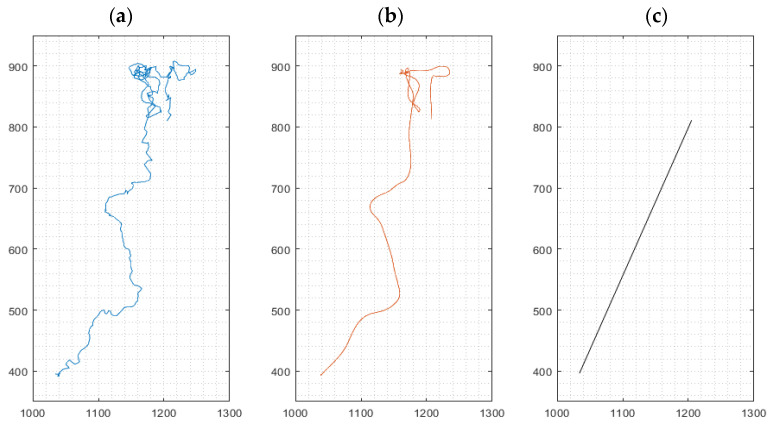
Representative path of a migrating cell during a 24 h experiment. Left: Path as a sequence of the cell’s centroids between frames (**a**). Middle: Smoothing of the path using robust linear regression (**b**). Right: Linear connection between the start and end point of the cell, indicating the Euclidian distance (**c**). *x*- and *y*-axis values indicate pixels (px); 1 px ≈ 1.15 µm.

**Figure 3 cells-13-00535-f003:**
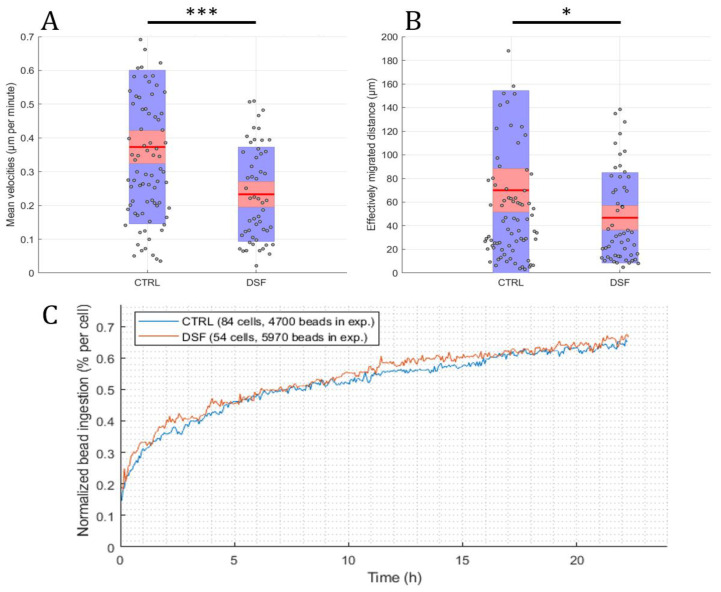
DSF decreased the cell migration and chemokinesis of cDC culture #20231. Phagocytes were enriched by four weeks of cell culture, then treated with DSF 50 nmol·L^−1^ or DMSO as a vehicle control (0.01% *v/v* final concentration) at 37 °C and 5% CO_2_ for 24 h and recorded by video microscopy using the Juli™ Br instrument. Before starting the video recording, cells were incubated with Dynabeads^®^ M-450 Epoxy at a cell to bead ratio of 1:20. (**A**) DSF decreased the cell migration speed (analyzed by smoothed path, as described in Figure 2b) of phagocytes derived from #20231. (**B**) DSF decreased the chemokinesis (analyzed by Euclidian distance, as described in Figure 2c) of phagocytes derived from #20231. Blue part of the bar: standard deviation; Red part of the bar: 95% confidence interval; Red line in the middle: mean. (**C**) DSF did not influence the phagocytosis of cells derived from #20231. Phagocytosis was performed by kinetic analysis for 24 h in a video frame: control (blue line), DSF 50 nM (orange line). Data are visualized as the mean from *n* = 84 cells from the control group and *n* = 54 cells from the DSF 50 nM group and evaluated by unpaired *t*-test (control vs. DSF 50 nM): * *p* < 0.05, *** *p* < 0.001. CTRL: Control; DSF: Disulfiram 50 nM.

**Figure 4 cells-13-00535-f004:**
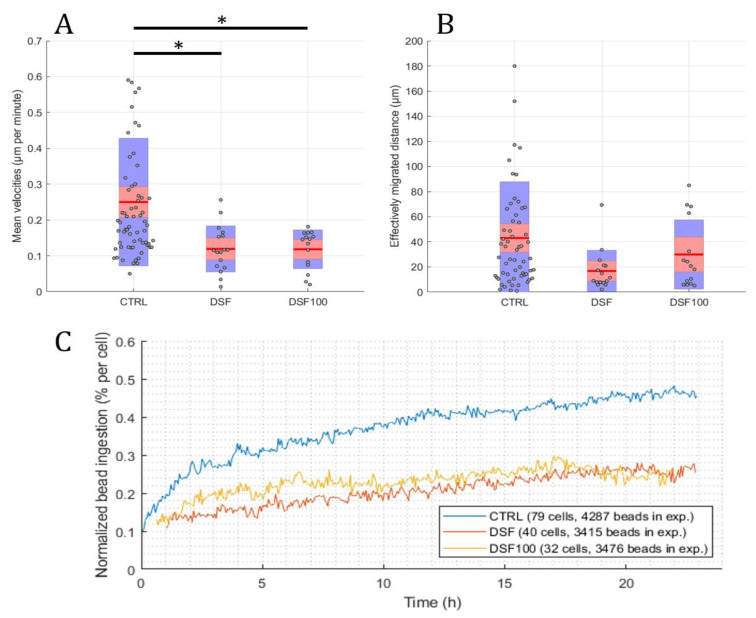
DSF decreased the cell migration speed, chemokinesis, and phagocytosis of iDC culture #20315. Phagocytes were enriched by four weeks of cell culture, then treated with DSF 50 nmol·L^−1^, 100 nmol·L^−1^ or DMSO as a vehicle control (0.01% *v/v* final concentration) at 37 °C and 5% CO_2_ for 24 h and recorded by video microscopy using the Juli™ Br instrument. Before starting the video recording, cells were incubated with Dynabeads^®^ M-450 Epoxy at a cell to bead ratio of 1:20. (**A**) DSF significantly decreased the cell migration speed (analyzed by smoothed path, as described in Figure 2b) of phagocytes derived from #20315. (**B**) DSF predisposed to decrease the chemokinesis (analyzed by Euclidian distance, as described in Figure 2c) of phagocytes derived from #20315 (no statistical difference). Blue part of the bar: standard deviation; Red part of the bar: 95% confidence interval; Red line in the middle: mean. (**C**) DSF decreased the phagocytosis of cells derived from #20315. Phagocytosis was performed by kinetic analysis for 24 h in a video frame: control (blue line), DSF 50 nM (orange line), DSF 100 nM (yellow line). Data are visualized as the mean from *n* = 58 cells from the control group, *n* = 16 cells from the DSF 50 nM group, and *n* = 15 cells from the 100 nM group and evaluated by unpaired *t*-test (control vs. DSF 50 nM or control vs. DSF 100 nM): * *p* < 0.05. CTRL: Control; DSF: Disulfiram 50 nM; DSF 100: Disulfiram 100 nM.

**Figure 5 cells-13-00535-f005:**
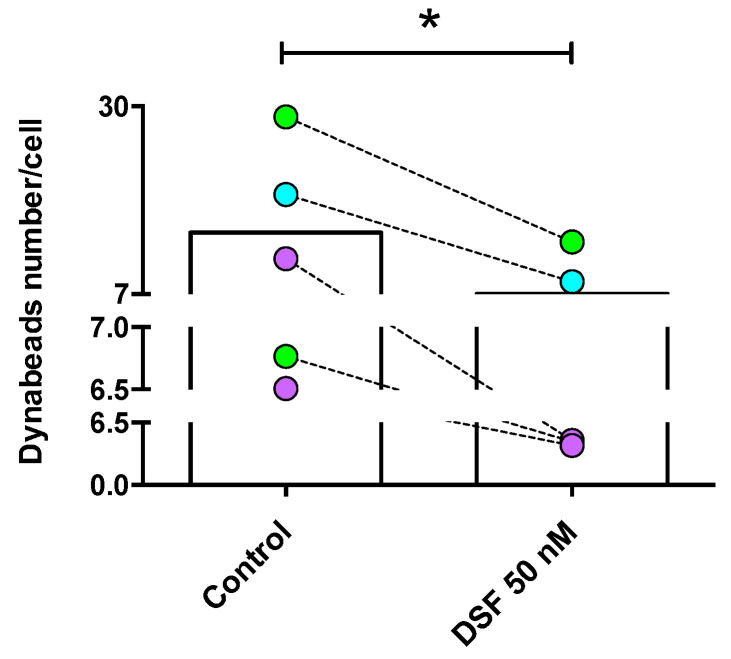
DSF-impaired phagocytosis of long-lived phagocytes. Phagocytes were cultured for 3–4 weeks, #20315-Glioblastoma (iDC) (turquoise-filled symbols), #20331-Glioblastoma (cDC) and #21161-Major depressive disorder (cDC) (light-green filled symbols), #20725-Borreliosis (M2a) and #21164-Post-coronavirus vaccination syndrome (M2a) (purple-filled symbols) were treated with DSF 50 nmol·L^−1^ or DMSO as a vehicle control (0.01% *v/v* final concentration) at 37 °C and 5% CO_2_ for 24 h, then incubated with Dynabeads^®^ M-450 Epoxy (20 beads/cell) for 24 h using the Juli™ Br instrument. Phagocytosis was assessed by counting the number of engulfed beads per cell by manual operation. The average total bead numbers engulfed per macrophage or respective dendritic cells with and without DSF treatment over 24 h. Data are visualized as the mean from *n* = 5 phagocyte cultures and evaluated by unpaired *t*-test (control vs. DSF 50 nM): * *p* < 0.05.

**Figure 6 cells-13-00535-f006:**
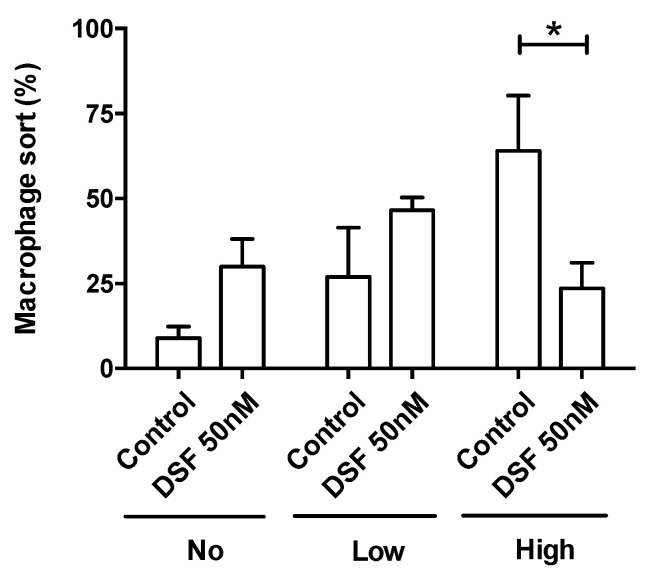
DSF decreased the rate of phagocytes with high phagocytic capacity. Phagocytosis was classified by the number of engulfed beads: (1) 0 beads, no capacity (No. of *x*-axis); (2) 1–10 beads, low capacity (Low of *x*-axis); and (3) >10 beads, high capacity (High of *x*-axis). Data are visualized as the mean ± SEM from *n* = 5, the same phagocyte cultures as Figure 4 (#20315, #20331, #20725, #21161, #21164) and evaluated by unpaired *t*-test (control vs. DSF): * *p* < 0.05.

**Figure 7 cells-13-00535-f007:**
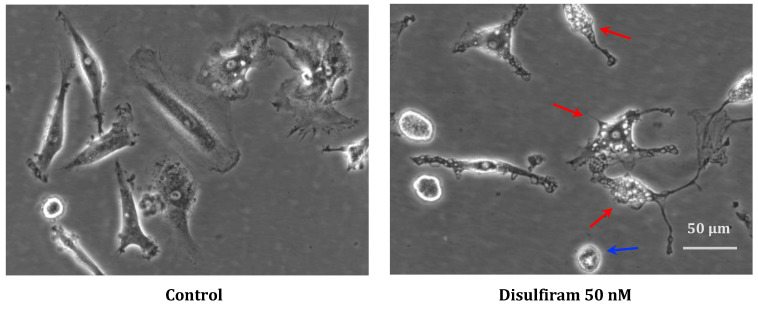
Representative morphology of long-lived phagocytes before and after DSF treatment. In vitro cultured phagocytes derived from patient GBM #20315 were incubated with DMSO as a vehicle control (0.01% *v/v* final concentration) (**left**), and morphological changes were observed after DSF 50 nmol·L^−1^ treatment for 24 h (**right**). The DSF-treated phagocytes displayed vacuolation (red arrow) and cell shrinkage (blue arrow), which was absent in control cultures. Images were phase-contrast images at a 20× objective, Nikon Eclipse Ts2 microscope. Scale bar: 50 μm.

**Figure 8 cells-13-00535-f008:**
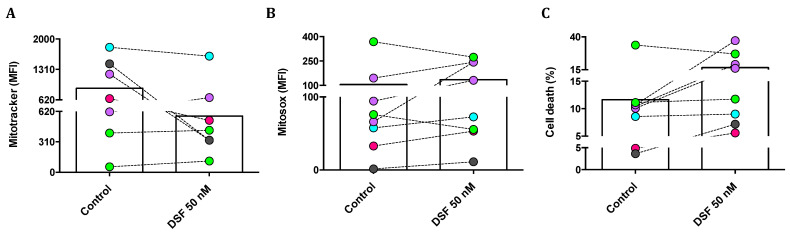
DSF-induced cell death by mitochondrial oxidative stress in macrophages but not in dendritic cells. Phagocytes were cultured for 3–4 weeks, then treated with DSF 50 nmol·L^−1^ or DMSO as a vehicle control (0.01% *v*/*v* final concentration) for 24 h at 37 °C and 5% CO_2_. Cells were analyzed by flow cytometry and mean fluorescence intensity (MFI) was quantified for mitochondrial mass of phagocytes detected by Mitotracker staining (**A**), mitochondrial ROS of phagocytes detected by Mitosox staining (**B**), and the percentage of dead cells were determined by PI staining (**C**). Phagocytes were identified as cDC from a CFS and an MDD patient (light-green symbols), iDC from an MND patient (turquoise-filled symbols), M1 macrophages from a healthy donor (grey-filled symbols), M2a macrophages from three patients with COVID, post-Vac, and long-COVID (purple-filled symbols), and M2c macrophages from another long-COVID patient (roseo-filled symbol). Accordingly, three macrophage cultures with an M2a phenotype and one with an M1 phenotype responded with increased mitochondrial ROS and cell death, whereas the mitochondrial ROS and viability of dendritic cells (iDC and cDC) were not significantly affected by DSF. cDC: conventional dendritic cells; iDC: immature dendritic cells; CFS: chronic fatigue syndrome; COVID-19: coronavirus disease; MDD: major depressive disorder; MND: motor neuron disease; VAC: vaccine.

**Figure 9 cells-13-00535-f009:**
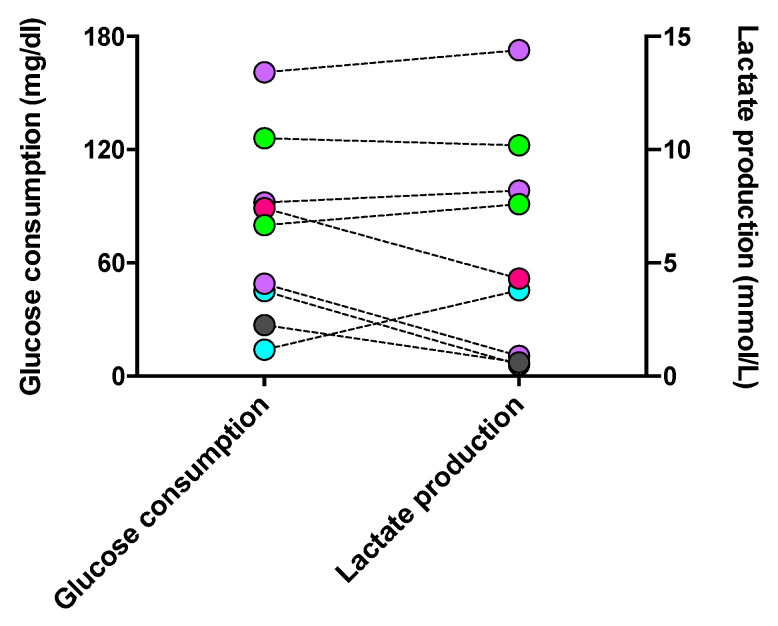
Glucose consumption and lactate production in long-lived phagocyte cultures. The supernatant of individual phagocyte cultures was tested for its contents in glucose and lactate, compared to the concentrations of the culture medium (10% FCS, IMDM). The increment of glucose concentrations between the medium and the glucose content of a given culture (left *y*-axis) and the corresponding lactate concentration (right *y*-axis) are shown for individual phagocyte cultures with color-coded phenotypes: cDC from a CFS and an MDD patient (light-green symbols), iDC from an MND patient (turquoise-filled symbols), M1 macrophages from a healthy donor (grey-filled symbols), M2a macrophages from three patients with COVID, post-Vac, and long-COVID (purple-filled symbols), and M2c macrophages from another long-COVID patient (roseo-filled symbol). CFS: chronic fatigue syndrome; COVID-19: coronavirus disease; GBM: glioblastoma; MDD: major depressive disorder; MND: motor neuron disease; VAC: vaccine.

**Figure 10 cells-13-00535-f010:**
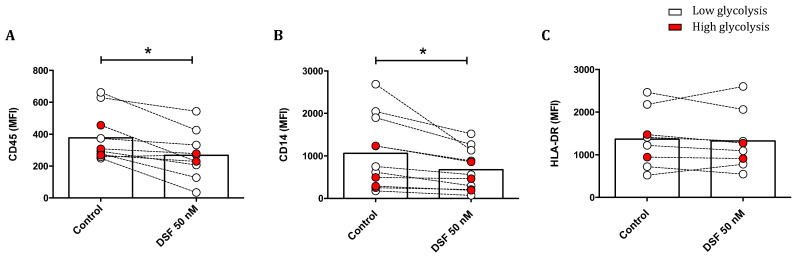
Modulation of surface antigens in phagocyte cultures differing in their glycolytic profile by DSF. Phagocytes were cultured for 3–4 weeks and treated with DSF 50 nmol·L^−1^ or DMSO as a vehicle control (0.01% *v/v* final concentration) for 24 h at 37 °C and 5% CO_2_. The mean fluorescence intensity (MFI) of leukocyte common antigen CD45 (**A**), monocyte–macrophage lineage marker CD14 (**B**), and MHC class II cell surface receptor HLA-DR (**C**) was examined using flow cytometry. The expression of CD45 and CD14 of general phagocytes especially in cells with metabolically low glycolysis was significantly decreased by DSF. Open circles signify phagocytes with low lactate metabolism; red-filled circles signify phagocytes with high lactate metabolism. Data are visualized as the mean from *n* ≥ 8 phagocyte cultures and evaluated by unpaired *t*-test (control vs. DSF 50 nM): * *p* < 0.05.

**Table 1 cells-13-00535-t001:** Overview of donors giving rise to long-lived phagocyte cultures.

	Diagnosis	Age	Gender	Phenotype	Exosome *
20231	Xanthogranuloma	43	M	80% cDC, 20% M1	21%
20315	Glioblastoma °IV	73	M	iDC	21%
20331	Glioblastoma °IV	66	F	cDC	16%
20365	Pancreatic cancer	55	M	M2a	26%
20373	COVID-19 infection after stem cell transplantation	18	M	M2a	10%
20643	Healthy donor	33	M	M1	5%
20650	Motor neuron disease	58	M	25% M1 + 75% iDC	1%
20667	Glioblastoma °IV	40	M	50% M1 + 50% iDC	2%
20705	Long-COVID	20	F	M2a	3%
20725	Borreliosis after DSF treatment	48	M	M2a	2%
21161	Major depressive disorder	30	F	cDC	10%
21162	Long-COVID	32	M	50% M1 + 50% cDC	4%
21163	Chronic fatigue syndrome	44	M	cDC	17%
21164	Post VAC syndrome	43	F	M2a	22%
21428	Prostate cancer	63	M	DC-10	7%

M = Male; F = Female; VAC = Vaccine. * The percentage of exosomes in a given flow cytometric analysis is denoted as the percentage of events corresponding to nucleated CD45-positive cells.

**Table 2 cells-13-00535-t002:** Immune phenotype of long-lived phagocytes.

PatNo.	CD14 [%postives]	CD206 [%postives]	CD163 [%postives]	CD38 [%postives]	CD141 [%postives]	Kv1.3 [%postives]
20231 *	20	2	3	6	6	7
20315 *	30	4	0.4	1	2	5
20331	83	5	1	1	2	27
20365	89	1	1	4	nt	7
20373	93	32	65	6	nt	76
20643	89	3	2	85	nt	nt
20650	88	0.3	0.2	25	nt	10
20667	98	3	2	52	nt	17
20705	99	nt	61	48	nt	78
20725	97	4	2	20	nt	53
21161	86	10	1	18	nt	nt
21162	98	nt	4	50	nt	nt
21163	63	nt	1	4	nt	nt
21164	78	nt	1	2	nt	nt
21428	90	54	70	nt	90	23

* Selective phagocyte cultures used to exemplify the analysis of phagocytosis and migration. nt: no data.

## Data Availability

Data are contained within the article and Appendix A.

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
