# Peer review of "Disulfiram Inhibits Opsonin-Independent Phagocytosis and Migration of Human Long-Lived In Vitro Cultured Phagocytes from Multiple Inflammatory Diseases"

_cells, 2024, doi:10.3390/cells13060535_

Round 1

Reviewer 1 Report

Comments and Suggestions for Authors

This study analyzed opsonin-independent phagocytosis and metabolic state of human monocyte-derived macrophages and dendritic cells, which were generated using monocytes from different types of patients. Phenotypic analysis of immune cells in different diseases is crucial to gain knowledge pathological mechanisms and establish treatment strategy. However, it is not easy to find the novelty of this manuscript. In addition, this reviewer concerns following points:

(1) Background of donors are too broad: i.e. n=1 Borreliosis, n=1 chronic fatigue syndrome, n=1 motor 81 neuron disease, n=1 pancreatic cancer, n=1 major depressive disorder, n=1 post corona-82 virus vaccination, n=3 coronavirus disease, and n=3 glioblastoma. Pathological mechanisms of these diseases are different. Why the authors had to use clinical samples from these donors, instead of using monocytes from healthy subjects, which are also possible to generate macrophages and DCs, if the freshly isolated cells were not used for analysis.

(2) Title: Please rephrase it. This study analyzed only monocyte-derived cultured macrophages and DCs, which were generated after four weeks of culture. Therefore, the use of a word “in vivo” is not appropriate.

(3) Please describe culture condition more in detailed. Which cytokines were used for culture of monocytes?

(4) An aim of this study was to assess migration behavior of cultured macrophages, but the assessment system did not include gradient of a chemoattractant. Thus only chemokinesis, but not chemotaxis (migration) can be assessed. Please rephrase a word “migration”.

(5) Table 1: It would be important to show FACS data to define the cell phenotype, and the frequencies of all cDC, M1, and M2a in the cultured cells from the patients.

(6) DSF 50 nM is relatively high. How was the concentration determined? There are many molecules inducing mitochondria oxidative stress and cell death. Why the effect of DSF had to be analyzed particularly?

(7) There are multiple methods to assess phagocytosis and migration. Please explain the advantage of the methods introduced in this manuscript.

(8) It would be better to generate cDC, M1, and M2a cells from healthy donors and assess whether function and reactivity of these cells were similar to those from patients.

Reviewer 2 Report

Comments and Suggestions for Authors

In this work Li and collaborators evaluate the effects of Disulfiram over distinct physiological features of human macrophages and dendritic cells. The authors design protocols in which use digital imaging processing to evaluate phagocytosis and migration of cells derived from distinct patients exhibiting different pathologies.

The significance of the study is the setting of methods or the evaluation of Disulfiram effects in these immune subsets. In this case the potential use or therapeutic significance of Disulfiram is not clear. The inclusion of other agent or agents known to interfere with phagocytosis could had been included to drag more relevant conclusions.

The work has nonetheless some flaws. As described it is not clear if the main aim of the work is the setting of such protocols or the evaluation of Disulfiram effects on the cell types described. The use of disulfiram as therapeutic drug in immune functions is not well stablished, and as described here it is not clear if just its use is instrumental to evaluate the protocols. The results depicted show only the data for some of the samples, and since the authors use very heterogenous populations and samples it is not clear if their conclusions can be applied in general to the cell types studied. There are also some typos and wrong labelling in some figures.

 Some points are listed below:

The authors used enriched monocytes that differentiated along long time. The sample for each patient is then very heterogenous. Using samples from healthy donors that could then differentiate to either macrophages or dendritic cells could had been a more simple and easy way to record the physiological parameters and have a “clean” effect and/or measurement. The high heterogeneity of their samples is cleared showed in the table 1.

To validate their digital processing (for phagocytosis) the authors could have compare with cytometric measurements. Have they performed this comparison?

Figure 6. It is not clear from the labelling and from the figure legend which sample (s) are showed here.

Comments on the Quality of English Language

In this work Li and collaborators evaluate the effects of Disulfiram over distinct physiological features of human macrophages and dendritic cells. The authors design protocols in which use digital imaging processing to evaluate phagocytosis and migration of cells derived from distinct patients exhibiting different pathologies.

The significance of the study is the setting of methods or the evaluation of Disulfiram effects in these immune subsets. In this case the potential use or therapeutic significance of Disulfiram is not clear. The inclusion of other agent or agents known to interfere with phagocytosis could had been included to drag more relevant conclusions.

Round 2

Reviewer 1 Report

Comments and Suggestions for Authors

The quality of manuscript was improved by the revision.

Author Response

Thanks very much for your support and confirmation. I will resubmit the final manuscript by checking all the typos and small mistakes.

Reviewer 2 Report

Comments and Suggestions for Authors

The new manuscript version has improved and satisfied my previous concerns. 

Author Response

(The authors gave the same response as above.)
